# Antioxidant and Inflammatory Gene Expression Profiles of Bovine Peripheral Blood Mononuclear Cells in Response to *Arthrospira platensis* before and after LPS Challenge

**DOI:** 10.3390/antiox10050814

**Published:** 2021-05-20

**Authors:** Magdalena Keller, Elisa Manzocchi, Deborah Rentsch, Rosamaria Lugarà, Katrin Giller

**Affiliations:** Animal Nutrition, ETH Zurich, Universitaetstrasse 2, 8092 Zurich, Switzerland; magdalena.keller@usys.ethz.ch (M.K.); elisa.manzocchi@usys.ethz.ch (E.M.); deborah.rentsch@usys.ethz.ch (D.R.); rosamaria.lugara@usys.ethz.ch (R.L.)

**Keywords:** microalgae, spirulina, inflammation, lipopolysaccharide, dairy cows, fattening bulls, leukocytes

## Abstract

Oxidative stress and inflammatory diseases are closely related processes that need to be controlled to ensure the desirable high performance of livestock. The microalga spirulina has shown antioxidant and anti-inflammatory properties in monogastric species. To investigate potential beneficial effects in ruminants, we replaced soybean meal (SOY) in the diets of dairy cows and fattening bulls by spirulina (SPI) and analyzed plasma concentrations of antioxidants (β-carotene, α-tocopherol, polyphenols) and serum total antioxidant capacity. Following in vitro stimulation with lipopolysaccharide (LPS), peripheral blood mononuclear cells (PBMCs) were isolated for expression analysis of inflammation- and antioxidant-defense-related genes. Plasma β-carotene concentration was higher in SPI, compared to SOY cows, but did not differ in bulls. Plasma total phenol concentration was significantly higher in SPI, compared to SOY bulls, but not in cows. Stimulation of bovine PBMCs with LPS increased the expression of most cytokines and some antioxidant enzymes. Gene expression of PBMCs derived from SPI animals, compared to SOY animals, hardly differed. Our results indicate that in ruminants, spirulina might not have potent antioxidant and anti-inflammatory properties. Future studies should evaluate the microbial degradation of spirulina and its bioactive compounds in the rumen to provide further data on potential beneficial health effects in ruminants.

## 1. Introduction

Oxidative stress and inflammatory diseases are common problems in high-yielding livestock and impair their health and productivity, thus resulting in economic losses for the farmers [1]. Dairy cows are most susceptible to oxidative stress and inflammation during the transition from pregnancy to lactation, resulting from substantial metabolic and physiological adaptations [1,2]. This contributes to the occurrence of inflammatory diseases such as mastitis and metritis. In fattening cattle, one of the major inflammatory disorders is bovine respiratory disease, recognized as the leading cause of death in feedlots in the USA [3]. Consequently, understanding and preventing oxidative stress and the associated inflammatory processes in livestock is of utmost importance.

Oxidative stress is characterized by an imbalance of reactive oxygen species (ROS) and antioxidants, including non-enzymatic antioxidants (e.g., carotenoids, tocopherols, polyphenols, and glutathione) and antioxidant enzymes (e.g., superoxide dismutase (SOD), catalase (CAT), glutathione peroxidase (GPX)) [4]. The accumulation of ROS can severely damage cellular macromolecules and continuously promote ROS production. Literature provides evidence for a vicious cycle with oxidative stress resulting in compromised inflammatory reactions, which, in turn, result in additional ROS production at the site of inflammation [5,6,7,8]. Inflammatory processes involve the actions of peripheral blood mononuclear cells (PBMCs), which comprise lymphocytes, natural killer (NK) cells, and monocytes, with the latter being able to differentiate to macrophages upon activation. Stimulation of the Toll-like receptor 4 (TLR4) on the surface of PBMCs by the bacterial endotoxin lipopolysaccharide (LPS) activates the transcription factor nuclear factor kappa B (NFKB). After translocation into the nucleus, NFKB induces the transcription of cytokines and chemokines such as tumor necrosis factor alpha (*TNFA*), interleukin 1 beta (*IL1**B*), and *IL8* (also known as C-X-C motif chemokine ligand 8 (*CXCL8*)). The proinflammatory cytokines TNFA and IL1B are important in the early onset of inflammation and defense. They direct leucocytes to the site of inflammation and can also cause systemic effects. Production of large amounts of TNFA can damage the organism and lead to a septic shock. Interferon gamma (IFNG) is produced by T-cells and NK cells and activates macrophages, which release ROS and thus contribute to oxidative stress at the site of inflammation [9]. The induction of prostaglandin–endoperoxide synthase 2 (PTGS2), also known as cyclooxygenase 2, leads to prostaglandin synthesis at the site of inflammation, which further modulates the inflammatory process [10]. In a negative feedback manner, activated macrophages produce IL10. It inhibits the activated macrophages and thus counteracts the inflammatory process, also by inhibiting cytokine production.

The close relationship between oxidative stress and inflammation has been described in the literature [11,12] and is further emphasized by the concomitant occurrence of oxidative stress and inflammatory production diseases such as mastitis and metritis in dairy cows during the periparturient period [13]. In contrast, antioxidant supplementation has been associated with a decreased incidence of mastitis [14,15]. Thus, to control common production diseases in livestock, both inflammatory processes and oxidative stress need to be addressed. In order to limit the globally increasing application of antibiotics in livestock production [16], it is desirable to find alternatives to antibiotics that can be applied both for treatment and prevention of production diseases. Such alternatives are searched for, among others, in the group of bioactive feeds containing compounds that exhibit antioxidant and anti-inflammatory properties and might therefore be advantageous for the prevention of oxidative stress and inflammation in livestock [17].

The filamentous blue-green microalga *Arthrospira platensis* (spirulina) belongs to the group of cyanobacteria and occurs naturally in alkaline lakes [18]. Spirulina is known for its high protein content (60–70%), which resulted in its consideration as a valuable protein source in livestock nutrition to replace the often critically discussed soybean meal [19,20]. This applicability has already been confirmed in different studies with ruminants that showed a maintained or even improved performance when soybean meal was replaced by spirulina [21,22,23,24,25]. In addition to its recognition as a sustainable protein source, spirulina has recently gained attention as a source of health-promoting bioactive compounds. These include carotenoids, tocopherols, polyphenols, γ-linoleic acid (GLA), and phycocyanin, all contributing to the observed antioxidant and anti-inflammatory effects of spirulina (reviewed by [26]). Spirulina has been shown to induce the activity of antioxidant enzymes such as SOD, GPx, and glutathione-S-transferase (GST), increase glutathione levels, decrease lipid oxidation, and consequently reduce oxidative stress in humans, rats, chickens, and fish [27,28,29,30,31]. Several studies have also reported an immunomodulatory effect of spirulina. Spirulina increased the accumulation of NK cells in tissues, induced phagocytosis by macrophages, modulated the production of key inflammatory molecules such as IL1B and IFNG in chickens, humans, and murine macrophages [32,33,34]. A close correlation has been reported between the antioxidant and anti-inflammatory activities of spirulina [35]. The proposed underlying mechanism of action is the inhibition of NFKB activation, a pivotal mediator of inflammatory processes, by spirulina’s bioactive compounds [36,37].

The antioxidant and anti-inflammatory properties of spirulina have been described in monogastrics such as rodents, humans, and chickens [28,33,38,39]. However, to the best of our knowledge, the interplay of spirulina, oxidative stress, and inflammation has not yet been investigated in ruminants. Since spirulina and its bioactive compounds may be significantly affected by the microbial metabolism in the rumen, the effects on oxidative stress and inflammation in ruminants might differ from those observed in monogastrics. This is especially interesting because spirulina is evaluated as an alternative protein source to replace soy for improved sustainability of both dairy cow and beef cattle nutrition and might be applied in large quantities in ruminant diets in the future.

Consequently, the aim of the present study was to assess a potential effect of dietary spirulina, compared to soybean meal, on basal and LPS-induced inflammation- and oxidative-stress-related gene expression of PBMCs obtained from dairy cows and fattening bulls. We hypothesized that (I) spirulina increases blood concentration of antioxidant molecules such as β-carotene, α-tocopherol, and polyphenols, as well as the total antioxidant capacity; (II) basal antioxidant and inflammatory gene expression is more beneficial in spirulina-fed, compared to soy-fed, animals; and (III) the gene expression response to in vitro challenge with increasing concentrations of LPS is less pronounced in spirulina-fed, compared to soy-fed, animals.

## 2. Materials and Methods

### 2.1. Experiment 1—Fattening Bulls

The experiment was approved by the Cantonal Veterinary Office of Zurich, Switzerland (license no. ZH129/18), and was conducted at the AgroVet-Strickhof research station (Lindau, Switzerland). A total of 12 Limousin-sired crossbred bulls with an initial age of 130 ± 8 (mean ± SD) days and an average body weight of 164 ± 9 kg were randomly divided into two experimental groups (*n* = 6 per group), balanced for initial body weight, sire, and breed of the dam (see also [40]). All animals were fed the same basal diet consisting of 50% grass silage, 30% maize silage, and 20% concentrate for five months. The main protein source in the concentrate differed between the two experimental groups: the control group received soybean meal (SOY), whereas the other group received the microalga spirulina (SPI) as part of the concentrate (Table 1). The spirulina had been cultivated in a Spirulina–Ogawa–Terui medium in production ponds [41]. After harvesting, vacuum belt filtering, and spray drying, the spirulina was sifted and blended (Institut für Getreideverarbeitung, Nuthetal, Germany). The crude protein contents of soybean meal and spirulina were 536 and 710 g/kg DM, respectively. The forage mixture was supplemented with 12 g/kg DM of a vitaminized mineral mix providing, per kg: 138 g Ca; 46 g P; 36 g Mg; 167 g Na; 5 g Zn; 3 g Mn; 1 g Cu; vitamin A, 625,000 IE, vitamin D_3_, 62,500 IE, vitamin E, 1 mg, and *Saccharomyces cerevisiae* (*NCYC Sc* 47), 333 colony-forming units. Additionally, animals had free access to a NaCl-containing licking block. The concentrates were formulated to be isocaloric and isonitrogenous. Diets were designed according to the Swiss feeding recommendations [42]. After five months of experimental feeding, blood samples were collected from the jugular vein after disinfecting the puncture site with 70% ethanol using serum, Na-heparinized, and EDTA plasma tubes, in that order, thereby flushing the needle before collecting the sample for PBMC isolation (Na-heparinized tube) to avoid contamination and activation of immune cells due to the skin perforation. The serum and EDTA plasma tubes were centrifuged at 2100× *g* for 10 min at 4 °C. The respective supernatants were collected and frozen at −80 °C until further analysis.

### 2.2. Experiment 2—Dairy Cows

The experiment was approved by the Cantonal Veterinary Office of Zurich, Switzerland (license no. ZH125/18), and was conducted at the AgroVet-Strickhof research station. A total of 12 multiparous, late-lactating dairy cows (Brown Swiss and Holstein) were randomly divided into two groups (*n* = 6 per group), balanced for breed, parity, days in milk, and milk yield (see also [24]). All animals were fed a hay-based diet with the addition of one of two isocaloric and isonitrogenous concentrates for 31 days. The control group received soybean meal (SOY), whereas the other group received the microalga spirulina (SPI) as part of the concentrate (Table 1). The spirulina was produced as described for experiment 1 but from a different batch. The crude protein contents of soybean meal and spirulina were 428 and 629 g/kg DM, respectively. In addition, the cows received 50 g/d of NaCL and 120 g/d of a vitaminized mineral mix. This mix contained per kilogram: 80 g Ca; 160 g P; 50 g Mg; 45 g Na; 4 g Zn; 2 g Mn; 0.5 g Cu; 0.02 g Se; 0.02 g I; 0.015 g Co, as well as, per kilogram, vitamin A, 1,200,000 IU; vitamin D3, 200,000 IU; vitamin E, 2 g. Diets were designed to cover requirements for maintenance and milk yield according to the Swiss feeding recommendations [42]. After 31 days of experimental feeding, blood samples were collected as described for experiment 1.

### 2.3. Plasma and Serum Analyses of Antioxidants and Total Antioxidant Capacity

The EDTA plasma samples were used to analyze the concentrations of β-carotene, α-tocopherol, and total phenols. For β-carotene and α-tocopherol, the plasma was deproteinized with pure water and ethanol (1:2) and extracted with hexane. For the analysis of β-carotene, the hexane was evaporated. The dried residue was dissolved in dichloromethane and dimethyl sulfoxide (1:4). The β-carotene content was analyzed with a UV/VIS detector and an HPLC system (Chromaster, Merck-Hitachi, Darmstadt, Germany) by applying the European standard method [43]. The α-tocopherol content was analyzed with a fluorescence detector and the same HPLC system according to Kälber et al. [44]. The total phenols from plasma were analyzed according to Sinz et al. [45]. Total antioxidant capacity (TAC) was measured in serum samples using the OxiSelectTM Total Antioxidant Capacity Assay Kit (Cell Biolabs, San Diego, CA, USA) according to the manufacturer’s protocol.

### 2.4. Short-Term Whole Blood Stimulation with LPS and Isolation of PBMCs

In the laboratory, the blood samples collected in Na-heparinized tubes in both experiments were carefully mixed by inversion. Per animal, six endotoxin-free cell culture tubes (VWR, Dietikon, Switzerland) were filled with 2 mL of blood (method adapted from [46]). The samples were treated with two different concentrations of LPS (purified LPS from Escherichia coli O111:B4; Merck, Buchs, Switzerland) in duplicate. As a control (LPS0), 20 µL of RPMI 1640 medium (Merck) were added to two tubes per animal. The remaining four tubes per animal were incubated in duplicate with 10 and 100 ng LPS/mL blood (LPS10 and LPS100), respectively. Dilutions of LPS were made in RPMI 1640 medium, and a total volume of 20 µL was added to the respective cell culture tubes. All tubes were incubated for 2 h in a water bath at 38 °C. After the incubation, all tubes were placed on ice to stop the reaction. Subsequently, samples were centrifuged (10 min, 1500× *g*, 4 °C), and the supernatant was removed. The red blood cells were lysed by resuspending the pellet in ice-cold ACK lysing buffer (Gibco, Thermo Fisher, CA, USA). Ice-cold phosphate-buffered saline (PBS) was added, and the remaining PBMCs were pelleted for 5 min at 300× *g* and 4 °C. After two more washings with ice-cold PBS and complete removal of the supernatant, the white cell pellets were frozen at −80 °C until further processing.

### 2.5. RNA Isolation, Quality Control, and cDNA Synthesis

To isolate total RNA from the PBMCs, TRIzol (Invitrogen, Thermo Fisher Scientific, Reinach, Switzerland) was added to the frozen cell pellets before thawing on ice according to the manufacturer’s protocol. Samples were homogenized and precipitated using chloroform. The RNeasy Micro Kit (Qiagen, Venlo, the Netherlands) was used to purify the RNA according to the manufacturer’s instructions. The RNA concentration was determined using a NanoDrop One (Thermo Fisher, MA, USA). The integrity of the RNA was determined using the Bioanalyzer (Agilent Technologies, CA, USA). The RNA integrity number (RIN) varied between 7 and 10. In total, 250 ng of RNA were used for reverse transcription with the GoScript™ Reverse Transcription System (Promega, Duebendorf, Switzerland) at the following conditions: 5 min at 25 °C, 60 min at 42 °C, 15 min at 70 °C. The cDNA was stored at −20 °C until further analysis. For two LPS0 samples from SOY dairy cows, not enough RNA was obtained, hence *n* = 4 in this group.

### 2.6. Gene Expression Analysis

The quantitative real-time polymerase chain reaction was performed using the KAPA Sybr Fast Mix (Kapa Biosystems, Wilmington, NC, USA) according to the manufacturer’s instructions on a LightCycler instrument (System LightCycler^®^ 96, Roche, Basel, Switzerland; used for inflammation-related genes) and a CFX384 Real-Time PCR Detection System (Bio-Rad, Munich, Germany; used for oxidative-stress related genes) with 40 cycles of amplification in three steps (10 s at 95 °C, 20 s at 60 °C, 1 s at 72 °C). The cycle of quantification (Cq) values resulted from applying a single threshold. Four reference genes were tested (actin beta (ACTB), H3.3 histone A (H3-3A), ubiquitin B (UBB), tyrosine 3-monooxygease/tryptophan 5-monooxygenase activation protein zeta (YWHAZ)). According to the geNorm output, the relative expression level (ΔCq) for each target gene was obtained by scaling the target gene Cq of each individual sample to the geometric mean of the Cqs of UBB and YWHAZ. Relative expression of target genes was related to the respective SOY LPS0 group. Primers (Table 2) were ordered at Microsynth (Balgach, Switzerland).

### 2.7. Statistical Analyses

The statistical analysis was performed using SPSS version 26 (SPSS GmbH Software, Munich, Germany) separately for both experiments. The Shapiro–Wilk test was used to test for normal distribution of data and residuals. Plasma and serum parameters were analyzed using Student’s *t*-test. Data analysis for qPCR results was performed on ΔCq values using a general linear model with diet group (SOY, SPI), LPS concentration (LPS0, LPS10, LPS100), and their interaction as fixed factors, while the individual animal was included as a categorical control variable. Differences in means were controlled for multiple comparisons using the Games–Howell post hoc test. Results are presented as boxplots of the relative gene expression with Spear style whiskers (min to max). Statistical significance was defined as *p* ≤ 0.05, and *p* ≤ 0.1 was considered a trend.

## 3. Results

### 3.1. Blood Concentrations of Antioxidants and Total Antioxidant Capacity

The plasma concentration of β-carotene was higher (*p* < 0.001) in SPI, compared to SOY dairy cows, whereas no difference was observed in fattening bulls (Figure 1A). In dairy cows, the concentrations of plasma total phenols and serum TAC did not significantly differ among SOY and SPI animals (Figure 1B,C). However, plasma total phenol concentration was higher (*p* = 0.045) and a trend (*p* = 0.074) was observed toward a higher serum TAC in SPI, compared to SOY bulls (Figure 1B,C). In both fattening bulls and dairy cows, plasma α-tocopherol was not affected by the diet.

### 3.2. Expression of Antioxidant Genes in Peripheral Blood Mononuclear Cells

Stimulation of PBMCs with LPS10 (*p* = 0.008), but not LPS100 (*p* = 0.077), decreased *GPX1* expression in SPI bulls, whereas no significant difference was observed in SOY bulls (Figure 2A). In dairy cows, *GPX1* expression did not significantly differ between groups (Figure 2F). For *CAT* expression, a pattern contrary to that of *GPX1* was observed, with no significant effect of diet and LPS in bull PBMCs (Figure 2B) and a decreased expression in PBMCs obtained from SPI (LPS100 versus LPS0, *p* = 0.049) but not from SOY cows (Figure 2G). In bull PBMCs, the expression of *SOD2* was increased in LPS10 and LPS100 (both *p* < 0.001), compared to LPS0 in both diet groups (Figure 2C). In PBMCs from dairy cows, a significant increase in *SOD2* expression was observed only in the SOY group (LPS10 versus LPS0, *p* = 0.049) since the observed increase in the SPI group (LPS100 versus LPS0, *p* = 0.071) could be considered only a trend (Figure 2H). The expression of NAD(P)H quinone oxidoreductase 1 (*NQO1*) was higher (*p* = 0.040) in SOY LPS100, compared to SOY LPS0 bull PBMCs, whereas LPS stimulation did not significantly affect *NQO1* expression in SPI bull PBMCs (Figure 2D). In dairy cow PBMCs, neither diet nor LPS stimulation affected *NQO1* expression (Figure 2I). The glutathione reductase (*GSR*) expression was lower (all *p* < 0.05) in PBMCs from bulls stimulated with LPS10 and LPS100, compared to LPS0 independent of diet group (Figure 2E). No significant difference in *GSR* expression was observed in PBMCs from dairy cows (Figure 2J). In PBMCs from both experiments, the mRNA expression of the transcription factor nuclear factor erythroid-derived 2 like 2 (*NFE2L2*) and the antioxidant enzyme SOD1 was not significantly affected by diet or LPS stimulation.

### 3.3. Expression of Inflammation-Related Genes in Peripheral Blood Mononuclear Cells

In PBMCs obtained from fattening bulls, the expression of *TLR4* was higher in LPS10 (*p* = 0.031) and LPS100 (*p* < 0.001), compared to LPS0 in SOY bulls, whereas no significant difference of *TLR4* expression was observed in SPI bulls (Figure 3A). The *TLR4* expression did not significantly differ in PBMCs from dairy cows in any of the diet groups (Figure 3J). The expression of *NF**KB2* was upregulated in LPS10 and LPS100 (all *p* < 0.01), compared to LPS0 PBMCs in bulls and cows independent of diet (Figure 3B,K). In fattening bulls, the mRNA expression of *TNF*, *IL1B*, *CXCL8*, *IL10*, and *PTGS2* was higher (all *p* < 0.01) for LPS10, compared to LPS0 PBMCs in both diet groups (Figure 3C–F,H). Stimulation with 100 ng/mL of LPS did not further increase the expression of any of these genes, as compared to 10 ng/mL of LPS. The same results (all *p* < 0.05) were observed in PBMCs from dairy cows (Figure 3L,M,O), except for *CXCL8* and *PTGS2*. The *CXCL8* expression did not significantly differ in PBMCs from dairy cows (Figure 3N). For *PTGS2*, an increase in expression in the SPI group was observed only with LPS100 (*p* = 0.003) but not with LPS10 (*p* = 0.137), compared to LPS0, whereas in the SOY group, the expression was higher in both LPS10 and LPS100, compared to LPS0 (Figure 3Q). The expression of *IFNG* in PBMCs from bulls was higher in SOY (*p* = 0.029) and SPI (*p* < 0.001) LPS100, compared to LPS0, whereas no increase was observed with LPS10 (Figure 3G). In cows, *IFNG* expression remained unaffected (Figure 3P). The BCL-2-associated X protein (*BAX*) expression was lower in LPS10 (*p* = 0.022) and LPS100 (*p* = 0.033), compared to LPS0 in SOY bulls, while in SPI bulls, no significant decrease in *BAX* expression was observed following LPS stimulation (Figure 3I). In cows, a contrasting effect was observed with a decrease in *BAX* expression in the SPI group with LPS10, compared to LPS0 (*p* = 0.036), and no significant effect (*p* = 0.169) in the SOY group (Figure 3R). In PBMCs from both bulls and cows, expression of the NFKB subunit *RELA* did not significantly differ due to diet or LPS stimulation.

## 4. Discussion

The microalga spirulina has been reported to possess antioxidant and anti-inflammatory properties in a wide range of in vivo studies performed in monogastric species [28,31,33,38,39]. These effects are in part attributed to spirulina’s contents of antioxidant and anti-inflammatory bioactive compounds including β-carotene, α-tocopherol, and polyphenols. Oral intake of spirulina has been demonstrated to increase plasma concentrations of β-carotene in rats [47]. In line with this, the β-carotene plasma concentration was almost doubled in SPI, compared to SOY dairy cows of the present study. This can be explained by the threefold higher β-carotene concentration in the SPI, compared to the SOY dairy cow diet (24.3 vs. 7.6 mg/kg DM). In contrast, no difference in plasma β-carotene concentration was observed in the fattening bulls. A potential explanation might be provided by the differing basal diets. The dairy cows received a basal diet consisting mainly of hay (74%), whereas the basal diet of the fattening bulls consisted of 50% grass silage. The concentration of β-carotene in the fattening bull diet was unfortunately not analyzed, but it is known that grass silage contains two- to threefold higher β-carotene proportions compared to hay [48]. Therefore, the β-carotene intake from the basal diet was likely higher in fattening bulls than in dairy cows. Additional dietary β-carotene from spirulina might thus have been absorbed more effectively in hay-fed dairy cows than in grass-silage-fed fattening bulls. This might also have been necessary since the excretion of β-carotene with the milk was higher in SPI, compared to SOY dairy cows [24], whereas fattening bulls do not experience this “loss” of β-carotene. Furthermore, other sex-specific effects on β-carotene metabolism might have played a role in the present findings since β-carotene metabolism is influenced by sex hormones [49]. The influence of culture conditions on spirulina composition can be ruled out in the present study because the spirulina used in both experiments was derived from the same batch. Consequently, also the differential effect of spirulina feeding on phenol plasma concentration seems to be related to the differing basal diet composition (similar total extractable phenol proportions in the dairy cow diets (SOY: 957 vs. SPI: 942 mg/100 g DM); not analyzed in fattening bull diets) and/or differences in bioavailability and metabolism of phenols between dairy cows and fattening bulls. Additionally, the time between the last intake of experimental feeds from the morning feed portion and blood sampling could have played a role since it was not controlled for and peak plasma levels of dietary phenols are reached after 1 to 2 h after ingestion and then rapidly decrease [50]. The observed trend for increased TAC in fattening bulls, accompanied by the significantly higher total phenol plasma concentration, is reasonably in line with studies suggesting that phenols are a major contributor to TAC [51,52]. Importantly, the different β-carotene plasma concentrations in dairy cows did not affect serum TAC. The similar plasma α-tocopherol concentrations in SPI and SOY animals is in line with the similar dietary proportions of α-tocopherol (SOY: 31.4 vs. SPI: 37.7 mg/kg DM in dairy cow diets; not analyzed in fattening bull diets) as well as previous results that did not observe differences in plasma α-tocopherol concentrations after supplementing rats with spirulina extracts [53].

Despite the observed differences in β-carotene and phenol plasma concentrations in SPI, compared to SOY dairy cows and fattening bulls, respectively, the expression of antioxidant enzymes hardly differed. This is in contrast with the literature that reports activation of NFE2L2 signaling by carotenoids and polyphenols, ultimately resulting in induced expression of antioxidant enzymes [54,55]. However, only a few studies regarding the potential antioxidant and anti-inflammatory effects of phenols in cattle are available to date, and their results are not consistent (summarized by [17]). Notably, the expression of the redox-sensitive transcription factor NFE2L2 and consequently that of its downstream antioxidant enzymes showed much higher variability in dairy cows, compared to fattening bulls, pointing toward larger interindividual differences, likely an effect of the varying metabolic activity during lactation. Studies with higher numbers of animals per group may be required to obtain more robust results. The immunomodulatory potential of spirulina with effects on the expression of pro-inflammatory molecules such as *IL1B*, *IL2*, *IL8*, *TNFA*, their transcription factor *NFKB*, as well as *PTGS2* in different monogastric species was summarized by Ravi et al. [56]. In vitro, incubation of human PBMCs with spirulina increased IFNG, IL1B, and IL4 secretion [33]. Mao et al. [34] observed a decrease in IL4 levels by spirulina in humans suffering from allergic rhinitis and oral intake of spirulina augmented the production of IFNG in humans [57]. In consequence, literature data on the direction of inflammation-related effects of spirulina are not consistent, but immunomodulatory effects have frequently been observed. In the present study, inflammation-related gene expression was hardly affected by spirulina supplementation in dairy cows and fattening bulls. The few differences observed were limited to an increased expression of *TLR4* in LPS10 and LPS100, compared to LPS0 treatments in SOY but not SPI bulls, as well as decreased expression of the proapoptotic *BAX* in LPS10-treated, compared to LPS0-treated, PBMCs in SPI but not SOY dairy cows. A numeric increase in *TLR4* expression with LPS stimulation was, however, also observed in SPI bulls and was apparently sufficient to significantly increase expression of *NFKB2* and several cytokines (*TNF*, *IL1B*, *CXCL8*, *IL10*, and *IFNG*). Due to the observed differences in β-carotene and phenol plasma concentrations, the similar expression of inflammation-related genes in SPI and SOY animals was surprising. Supplementation with β-carotene in vitro in murine macrophages and in vivo in mice prevented intracellular ROS accumulation and inhibited the expression of inflammatory genes such as *PTGS2*, *IL1B*, and *TNFA* upon LPS stimulation [36,58]. In dairy cows, Wang et al. [59] observed lower incidences of mastitis after supplementation with β-carotene. An underlying mechanism for the beneficial effects of β-carotene observed in monogastrics seems to be the blockage of the NFKB p65 subunit (RELA) nuclear translocation [36]. Since we analyzed only gene expression but not nuclear protein proportions of RELA, we cannot state if this mechanism was at all induced in our dairy cows. Additionally, the higher plasma phenol concentration in fattening bulls did not affect the expression of inflammation-related genes in isolated PBMCs.

Taken together, despite the broad evidence of antioxidant and immunomodulatory effects of spirulina in a variety of monogastric species, the present study could not confirm these effects on the level of antioxidant blood parameters and gene expression in PBMCs obtained from ruminants both in the native and the LPS-challenged state. Ruminants efficiently digest especially the carbohydrate fraction of spirulina, but about 20% of dietary spirulina still bypasses the rumen [60,61]. Currently, not much information exists about the rumen degradability of spirulina’s bioactive compounds during ruminal fermentation. It seems, however, likely that the different digestive processes in monogastrics and ruminants result in a differing bioavailability of spirulina’s bioactive compounds. This might, in turn, also differently affect the beneficial effects on oxidative and inflammatory processes in the animals’ metabolism. Our results indicate that in ruminants, plasma β-carotene, α-tocopherol, and phenols are likely not involved in mediating antioxidant and anti-inflammatory gene expression in PBMCs: first, some of these antioxidants were present in similar concentrations in plasma in both diet groups, and second, even when concentrations differed, expression of antioxidant and inflammatory genes was not significantly affected.

It has previously been suggested that the deep blue chromophore–protein complex phycocyanin, forming up to 40% of spirulina’s protein, may be the major antioxidant compound in spirulina [26,62]. Phycocyanin was reported to inhibit radical generation and lipid peroxidation in vitro and in vivo [35,63]. The chromophore phycocyanobilin was shown to possess an antioxidant activity very similar to that of phycocyanin, suggesting that phycocyanobilin is responsible for most of the antioxidant effect of phycocyanin [62]. However, a decrease in the antioxidant activity of phycocyanin after trypsin-mediated hydrolysis suggests that also the apoprotein significantly contributes to the antioxidant properties of phycocyanin [62]. In addition, phycocyanin possesses immunomodulatory properties by regulating cytokine expression and selectively inhibiting PTGS2, whereas results concerning anti-inflammatory effects following LPS-stimulation of macrophages remain contradictory [64,65,66]. The beneficial antioxidant effects of phycocyanin have been investigated so far only in monogastrics. Interestingly, lactic acid fermentation of spirulina has been shown to release phenols and phycocyanin and to improve the antioxidant activity in vitro [67]. This effect was, however, diminished when fermentation exceeded 36 h. It can thus be speculated that by hydrolyzing the cell wall of spirulina, rumen fermentation could in a first step enhance the release and thus the bioavailability of bioactive compounds including phycocyanin. However, prolonged exposure to fermentation processes presumably leads to microbial degradation of the protein compound of phycocyanin and potentially also the chromophore. Thus, the compound contributing significantly to spirulina’s antioxidant and immunomodulatory properties may be lost in ruminants.

Some additional aspects need to be considered when evaluating the present results, namely, (a) a direct comparison of the outcomes of the two experiments in the present study has to be regarded with care because of substantial differences in basal diets, duration, and proportion of spirulina supplementation, animal sex, age, and production stage. Further studies are required to directly compare the effects of spirulina in male and, preferably non-lactating, female animals of similar age; (b) the LPS concentrations applied in the present study were rather high. In most cases, the inflammatory response was not further increased by LPS100, compared to LPS10, as indicated by similar cytokine expression. Therefore, it can be assumed that LPS10 already induced the maximum inflammatory response. Presumably, the antioxidant and anti-inflammatory effects of spirulina were not potent enough to counteract the induction of the maximum inflammatory response. It can only be assumed that when using a lower concentration of LPS, representing a milder inflammatory state, potential attenuating effects of dietary spirulina intake on cytokine expression might have been observed. This should be further investigated in additional ex vivo studies; (c) in the case of the dairy cows, it needs to be considered that the animals were all in the late-lactating stage. Dairy cows are most susceptible to oxidative stress and inflammation during the transition period when substantial metabolic and physiological adaptations occur [1,2]. Therefore, it would be interesting to observe the effects when subjecting transition cows to spirulina supplementation; (d) the analysis of gene expression alone does not provide complete information about dietary effects. Instead of a change in gene expression, the activity, e.g., of antioxidant enzymes may be affected by spirulina as has been previously reported [30]. Due to the minor differences in serum TAC, an increase in antioxidant enzyme activity seems, however, unlikely; (e) as a consequence of intense hepatic xenobiotic metabolism, changes in gene expression might have been limited to liver tissue and hepatic immune cells and were consequently not visible in the circulating immune cells; (f) the type and concentration of LPS and the type of leukocytes play a role in the results [64,65,66]. Using PBMCs in the present study as well as a different type and concentration of LPS compared to other studies may have contributed to differing results; (g) interestingly, the antioxidant effect of spirulina seems to taper off when exceeding a specific threshold concentration [68]. This threshold, however, is not yet determined and might also differ between different species. The dietary proportion of 4–5% of spirulina in DM fed in the present study resulted in spirulina intakes of 0.8 g/kg body weight (fattening bulls) and 1.3 g/kg body weight (dairy cows). This proportion is slightly higher than what has been applied in studies with rats (0.5–1 g/kg body weight [28,69]) and much higher than what was applied in a study with humans (<0.1 g/kg body weight [30]). If spirulina’s antioxidant and anti-inflammatory properties become only active at a specific low intake level, as indicated by the results of Ding et al. [68], this might at least in part explain the absence of effects in the present study. Further studies should evaluate if such threshold concentrations exist, and if so, if and how they differ between different species and interact with rumen metabolism.

## 5. Conclusions

The effects of dietary spirulina on antioxidant plasma concentrations in ruminants differed in the experiments in fattening bulls and dairy cows of the present study. Our hypothesis I was mostly disproved since the only significant increases in blood antioxidants were observed for β-carotene (in dairy cows) and total phenol (in fattening bulls). Basal mRNA expression of antioxidant and inflammatory genes in PBMCs did not differ between the experimental groups, thus disproving hypothesis II. The partially higher plasma antioxidant concentrations observed after spirulina feeding were clearly not effective in attenuating the oxidative and inflammatory effect of increasing LPS concentrations, as demonstrated by similar changes in the expression of antioxidant enzymes and cytokines in PBMCs from SPI and SOY animals, which disproves hypothesis III. The results of the present study obtained from two independent feeding experiments indicate that in ruminants, spirulina might not have antioxidant and anti-inflammatory properties similar to those observed in monogastric species. Rumen degradation of key bioactive compounds such as phycocyanin might explain the species-related differences in spirulina-mediated effects, which requires further investigation. In conclusion, further research is required to understand the rumen degradation, bioavailability, metabolism, and actions of spirulina and its bioactive compounds in ruminants and to provide further data on potential beneficial health effects in fattening bulls and dairy cows.

## Figures and Tables

**Figure 1 antioxidants-10-00814-f001:**
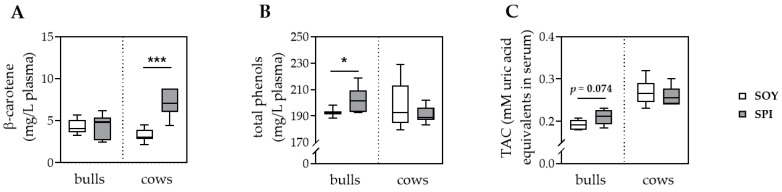
Antioxidants and antioxidant capacity in blood of fattening bulls and dairy cows fed with soybean meal (white boxes) or spirulina (gray boxes) as protein source. The concentrations of β-carotene (**A**) and total phenols (**B**) were analyzed in plasma, whereas the total antioxidant capacity (TAC, **C**) was determined in serum. Results are presented in boxplots with Spear style whiskers (min to max). * *p* < 0.05; *** *p* < 0.001.

**Figure 2 antioxidants-10-00814-f002:**
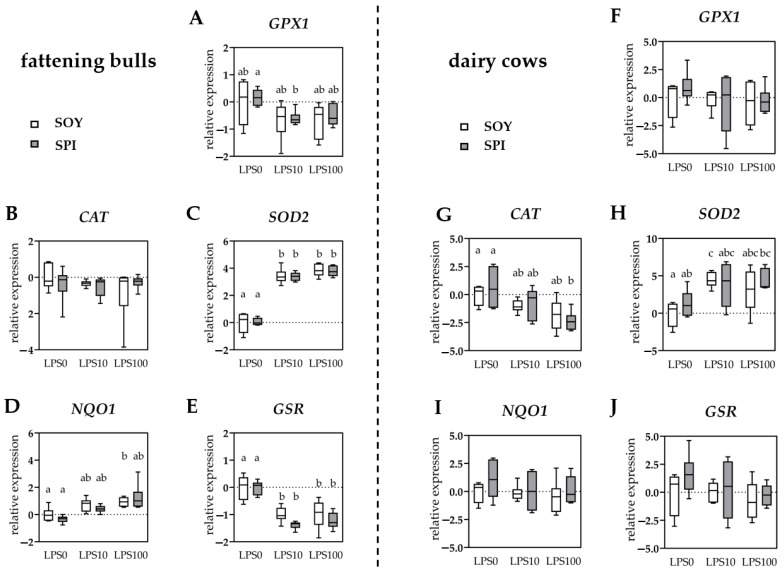
Relative expression of antioxidant genes in peripheral blood mononuclear cells isolated from fattening bulls (**A**–**E**) and dairy cows (**F**–**J**) fed with soybean meal (white boxes) or spirulina (gray boxes) as protein source. Individual results were normalized to the mean of the respective SOY LPS0 group. Results are presented in boxplots with Spear style whiskers (min to max). Different lowercase letters indicate significant differences (*p* ≤ 0.05).

**Figure 3 antioxidants-10-00814-f003:**
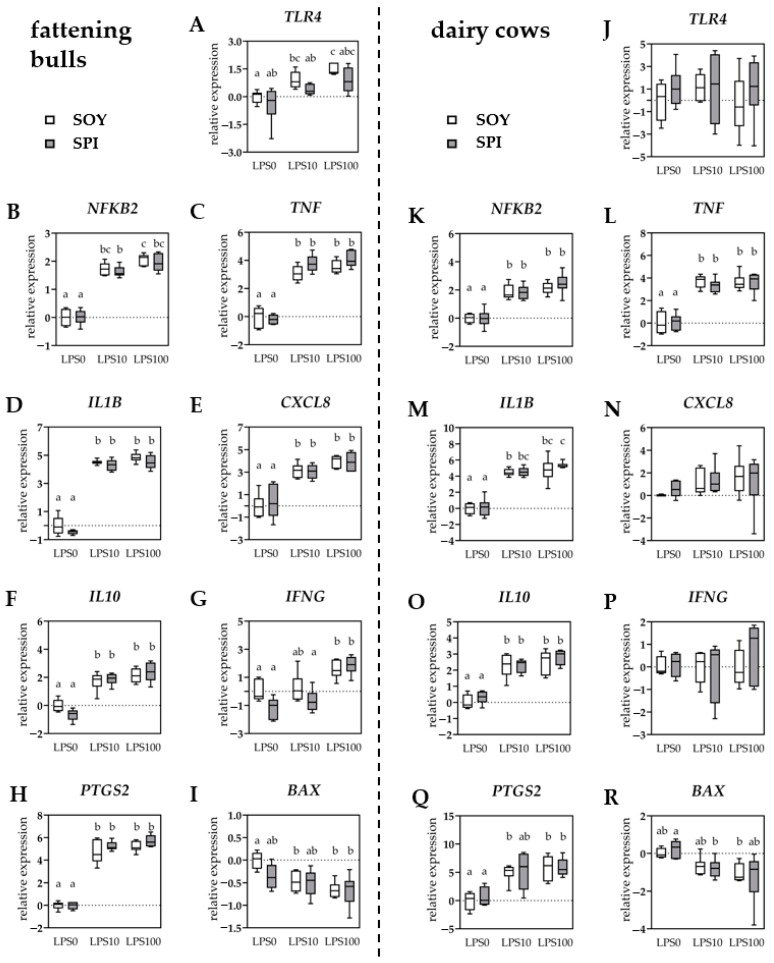
Relative expression of inflammation-related genes in peripheral blood mononuclear cells isolated from fattening bulls (**A**–**I**) and dairy cows (**J**–**R**) fed with soybean meal (white boxes) or spirulina (gray boxes) as protein source. Individual results were normalized to the mean of the respective SOY LPS0 group. Results are presented in boxplots with Spear style whiskers (min to max). Different lowercase letters indicate significant differences (*p* ≤ 0.05).

**Table 1 antioxidants-10-00814-t001:** Components of the experimental concentrate used in experiment 1 (fattening bulls) and the experimental total mixed ration used in experiment 2 (dairy cows), their chemical composition and contents of extractable phenols, β-carotene, and α-tocopherol.

	Experiment 1—Fattening Bulls (Concentrate)		Experiment 2—Dairy Cows (Total Mixed Ration)	
	Control	Spirulina	Control	Spirulina
Ingredients (g/kg DM)				
Hay	-	-	740	740
Sugar beet pulp	-	-	130	130
Soybean meal	277	-	60	-
Spirulina	-	198	-	50
Wheat flakes	-	-	40	50
Molasses	20	20	30	30
Wheat	450	518	-	-
Maize grain	237	250	-	-
Wheat bran	11	-	-	-
Tallow-lard mixture	5	5	-	-
Composition (%)				
DM (% of wet weight)	88.2	88.1	65.4	52.4
OM	96.8	97.1	85.6	86.1
CP	22.8	21.4	14.6	14.2
NDF	46.4	n.a.	46.9	43.7
ADF	10.5	n.a.	30.5	27.5
ADL	n.a.	n.a.	5.22	4.10
EE	2.60	3.71	1.64	2.36
Total extractable phenols (mg/100 g DM)	n.a.	n.a.	957	942
β-carotene (mg/kg DM)	n.a.	n.a.	7.6	24.3
α-tocopherol (mg/kg DM)	n.a.	n.a.	31.4	37.7

ADF: acid detergent fiber; ADL: acid detergent lignin; CP: crude protein; DM: dry matter; EE: ether extract; n.a.: not analyzed; NDF: neutral detergent fiber; OM: organic matter; -: not contained in the diet.

**Table 2 antioxidants-10-00814-t002:** Sequences of the primers used for qPCR in both experiments.

Gene	Accession Number	Forward Primer (5′–3′)	Reverse Primer (5′–3′)	Amplicon Length
*ACTB*	NM_173979.3	GAT CTG GCA CCA CAC CTT CT	AGA GAC AGC ACA GCC TGG AT	174
*BAX*	NM_173894.1	GCC CTT TTG CTT CAG GGT TT	ACA GCT GCG ATC ATC CTC TG	179
*CAT*	NM_001035386.2	CTG GGA CCC AAC TAT CTC CA	AAG TGG GTC CTG TGT TCC AG	179
*CXCL8*	NM_173925.2	GTT GCT CTC TTG GCA GCT TT	GGT GGA AAG GTG TGG AAT GT	118
*GPX1*	NM_174076.3	CCT GAC ATT GAA ACC CTG CT	TCA TGA GGA GCT GTG GTC TG	220
*GSR*	NM_001114190.2	TGT CAT TGT TGG TGC TGG TT	AGC GTT CTC CAG CTC TTC AG	154
*H3-3A*	NM_001014389.2	GTA CTG TGG CAC TCC GTG AA	GAT AGG CCT CAC TTG CCT CC	168
*IFNG*	NM_174086.1	TTC TTG AAT GGC AGC TCT GA	TTC TCT TCC GCT TTC TGA GG	154
*IL1B*	NM_174093.1	TGA CCT GAG GAG CAT CCT TT	AGA GGA GGT GGA GAG CCT TC	179
*IL10*	NM_174088.1	GTG AAC TCA CTG GGG GAG	ACC GCC TTG CTC TTG TTT T	92
*RELA*	NM_001080242.2	GCC TGT CCT CTC TCA CCC CAT CTT TG	ACA CCT CGA TGT CCT CTT TCT GCA CC	152
*NFKB2*	NM_001102101.1	ATC TGA GCA TTG TGC GAC TG	CTT CAG GTT TGA GGC TCC AG	131
*NQO1*	NM_001034535.1	ATG AAG GAG GCT GCC ATA GA	CTG CAG CTT CCA GCT TCT TT	223
*NFE2L2*	NM_001011678.2	CTC CAG CCA GTT GAC AGT GA	GTT GTG CTT TCA GGG TGG TT	225
*PTGS2*	NM_174445.2	TCG AGG TGT ATG TAT GAG TGT A	GTG CTG GGC AAA GAA TGC AA	487
*SOD1*	NM_174615.2	AGA GGC ATG TTG GAG ACC TG	CAG CGT TGC CAG TCT TTG TA	189
*SOD2*	NM_201527.2	ATT GCT GGA AGC CAT CAA AC	AGC AGG GGG ATA AGA CCT GT	192
*TLR4*	NM_174198.6	GAC CCT TGC GTA CAG GTT GT	GGT CCA GCA TCT TGG TTG AT	103
*TNF*	NM_173966.3	GCC CTC TGG TTC AAA CAC TC	AGA TGA GGT AAG CCC GTC A	191
*UBB*	NM_174133.2	AGA TCC AGG ATA AGG AAG GCA T	GCT CCA CCT CCA GGG TGA T	198
*YWHAZ*	NM_174814.2	AGG CTG AGC GAT ATG ATG AC	GAC CCT CCA AGA TGA CCT AC	140

## Data Availability

Data will be made available on request by the corresponding author for reasonable purposes.

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
