# Peer review of "Antioxidant and Inflammatory Gene Expression Profiles of Bovine Peripheral Blood Mononuclear Cells in Response to Arthrospira platensis before and after LPS Challenge"

_antioxidants, 2021, doi:10.3390/antiox10050814_

Round 1

Reviewer 1 Report

In this paper I reviewed the Authors evaluated the replacement of soybean meal in the diet of dairy cows and fattening bulls with spirulina by assessing animals plasma concentrations of many antioxidants compound (β-carotene, α-tocopherol, polyphenols) and serum total antioxidant capacity.

The paper reports interesting and novel findings and the topic investigated fits well the scope of Antioxidants journal.

The results have been properly reported as well as well compared with the available literature investigating similar topics.

However, the paper could be further improved by some changes as follows:

Lines 28-31: a couple of additional references may further support the statements;

Line 77: The following reference may be useful: Tufarelli et al. (2021). Effects of horsetail (Equisetum arvense) and spirulina (spirulina platensis) dietary supplementation on laying hens productivity and oxidative status. Animals11(2), 335.

Line 112: It could be useful to add the reference used to determine the dietary animals' requirements.

Table 1: add the crude protein content (%) of the soybean meal included in diets.

Table 1: Did diets (concentrates or TMRs) contain a mineral-vitamin premix?

Lines 220-222: Delete (it was a mistake in formatting the paper)

Line 448: The conclusion section could be further improved.

Reviewer 2 Report

In the paper “Antioxidant and Inflammatory gene expression profiles of bovine peripheral blood mononuclear cells in response to Arthrospira Platensis before and after LPS challenge” the Author evaluated the effects of spirulina on the expression of several genes related to both antioxidant and inflammatory pathways on PBMC derived from fattening bull and dairy cow fed with spirulina supplemented diet.

Despite the paper is of some interest, some major criticisms should be addressed:

-             As the authors state in the discussion as “additional aspects” (Lines 411-413), the obtained results between bulls and cow should not never be compared throughout the text, because of differences in diet, period of administration, g/Kg DM of spirulina added, etc. For these reasons, related sentences should be removed.

-             The authors in the discussion (lines 414-422) hypothesize that the LPS concentrations were too high to be successfully counteracted by spirulina. The observation maybe is correct, for this reason additional “ex vivo” experiments should be performed using LPS less concentrated.

-             The Authors did not explain when the blood from animals has been collected after the last treatment; this issue can be crucial for the results obtained after LPS stimulation.

-             The results section should be simplified; the suggestion is to remove graphs and data related to non-significant results.

-             In lines 441-447 the authors explain the scarce efficacy of spirulina when supplemented to ruminants on the basis of the doses administered in rats and human beings (less than in ruminants) and concluding that spirulina exerts the anti-inflammatory effects only if administered in lower doses. This affirmation is speculative, the more reliable explanation is maybe related to the rumen fermentation leading to disruption of phycocyanin, as well discussed in lines 390-410.

Considering the previous issues, the paper can be published on Antioxidants only after major revision.

Minor remarks:

  • Line 167: the authors state that three different LPS concentrations were used, but only two have been added (LPS0 are control)
  • Line 173: why the incubation has been carried out at 38°C?
  • Line 243: change “between” as “among”

Round 2

Reviewer 2 Report

The authors have improved the paper and answered satisfactorily to the remarks.